# Factors Influencing Microbiota in Modulating Vaccine Immune Response: A Long Way to Go

**DOI:** 10.3390/vaccines11101609

**Published:** 2023-10-18

**Authors:** Francesca Romana Ponziani, Gaetano Coppola, Pierluigi Rio, Mario Caldarelli, Raffaele Borriello, Giovanni Gambassi, Antonio Gasbarrini, Rossella Cianci

**Affiliations:** Department of Translational Medicine and Surgery, Catholic University, Fondazione Policlinico Universitario A. Gemelli, IRCCS, 00168 Rome, Italygcoppp@gmail.com (G.C.); pierluigi.rio18@gmail.com (P.R.); mario.caldarelli01@icatt.it (M.C.); raffaeleborr@gmail.com (R.B.); giovanni.gambassi@unicatt.it (G.G.); antonio.gasbarrini@unicatt.it (A.G.)

**Keywords:** vaccine immunogenicity, microbiota, immune system

## Abstract

Vaccine immunogenicity still represents an unmet need in specific populations, such as people from developing countries and “edge populations”. Both intrinsic and extrinsic factors, such as the environment, age, and dietary habits, influence cellular and humoral immune responses. The human microbiota represents a potential key to understanding how these factors impact the immune response to vaccination, with its modulation being a potential step to address vaccine immunogenicity. The aim of this narrative review is to explore the intricate interactions between the microbiota and the immune system in response to vaccines, highlighting the state of the art in gut microbiota modulation as a novel therapeutic approach to enhancing vaccine immunogenicity and laying the foundation for future, more solid data for its translation to the clinical practice.

## 1. Introduction

Vaccination represents one of the medical procedures with the highest impacts on public health in the history of medicine; currently, available vaccines are expected to save 2–3 million lives annually [1]. Despite the improvements in the development of new vaccines, with many pathogens whose infection can now be prevented, the heterogeneity of vaccinal response among individuals still constitutes a challenge in preventive medicine and a critical issue in global health [2,3].

Among the factors that influence humoral and cellular responses to vaccines in humans, the most relevant are intrinsic host factors (i.e., age, genetics, sex, comorbidities, microbiota), perinatal influences (i.e., gestational age, birth weight, feeding method), extrinsic influences (i.e., infections, antibiotics), environmental elements (i.e., geographic location, season), behavioral aspects (i.e., smoking, alcohol consumption, exercise), and nutritional habits. In addition, the vaccine type, dose, and adjuvants, as well as administration factors, such as the schedule, site, and time of vaccination, are also determinants [4].

The heterogeneity of vaccine responses is therefore linked to a complex network of interactions that ultimately results in the different immunoreactivity of specific populations, such as infants [5], elderlies [3,6], frail individuals in an immunodeficient state due to chronic diseases or malnutrition [3,7], or individuals from specific ethnicities or with determined social behaviors [8,9,10].

Several age- and disease-related alterations in the immune system (such as reduced production of antibodies and memory B cells in infants or immune modifications typical of immunosenescence in older adults) [5,6] could partially explain this heterogeneity. In this scenario, the human microbiota represents one of the factors with a significant influence on vaccinal immunity and individual determinants [7] and is a key to understanding and potentially modifying this heterogeneity. Indeed, the human microbiota, which comprises more than one thousand microorganisms, including bacteria, fungi, and protozoa dwelling on mucosal surfaces and the skin, is a pivotal factor in the balance of the systemic inflammatory status [7,11] and a modulator of the immune response. To further enrich this field, it is worth mentioning that mental health and psychological disorders can largely affect the immune system, also through the gut microbiota (GM); indeed, the increased sympathetic tone in the gut can affect acid secretion, bile acid production, and intestinal peristalsis, resulting in reduced bacterial clearance, alterations in the gut microbiota composition, and small intestinal bacterial overgrowth (SIBO) [12,13]; in addition, enteric nerve plexus activity can interfere with phagocytosis and diapedesis, which are critical for gut immune homeostasis [11,14]. However, no specific evidence on the possible role of vaccines is available. The human microbiota, particularly the gut microbiota, interacts with the immune system in different ways, leading to either proinflammatory or anti-inflammatory activation [7,11]. This interplay takes place through various mechanisms, such as the production of bacterial metabolites with pro-/anti-inflammatory activity, the regulation of immune cell differentiation, the influence on the host’s metabolism, and the modulation of intestinal permeability [7,10,15]. All of these factors can result in an imbalance when external or internal factors perturb the equilibrium in the GM community, inducing an impairment in its functional and compositional harmony called gut dysbiosis [16]. Gut dysbiosis has been found to be involved in the pathogenesis of several conditions, including gastrointestinal, endocrine, cardiovascular, and metabolic diseases [10], and the dysregulation in the systemic inflammatory status associated with this condition can act as a trigger for a dysfunctional or impaired response to immune stimulations, including vaccinal immunity [7]. Thus, the GM represents a potential key factor in modulating vaccine efficacy [7], with possible future implications for the use of GM modulation in this field (Figure 1).

This narrative review aims to analyze the relationship between GM perturbations and their consequences on the immune response to vaccines, highlighting the state of the art in GM modulation as a novel therapeutic approach to enhance vaccine immunogenicity.

## 2. Microbiota and Immune Modulation

The human microbiota consists of thousands of different species, and its characterization is continuously improving thanks to new “omics” technologies [17,18,19]; moreover, the microbiota composition also shows a certain degree of inter- and intraindividual variability based on a variety of environmental factors and the genetic background [20,21,22]. Differences have also been reported for the microbiota of different body sites.

### 2.1. Gut Microbiota and Immune Response

Most efforts to understand the immunomodulating properties of the microbiota have targeted the gut. The first evidence of the molecular mechanisms underlying the close interaction between the GM and the immune system dates back to the 2000s; in 2004, Rakoff-Nahoum et al. showed that Toll-like receptors (TLRs), recognizing the microbial products of both pathogens and commensal bacteria, could be considered at least one of the many links involved in the GM–host interaction [23]. Similarly, in 2008, it was shown that a bacterial strain normally part of the GM in mice (*Bifidobacterium infantis* 356624) could affect the activation of the nuclear factor kappa-light-chain-enhancer of activated B cells (NF-kB), which express integrin α4β7 and chemokine receptor 9 (CCR9) on their surfaces [24], and of T regulatory (Treg) cells [25], thereby paving the way for the novel concept of GM-mediated modulation of the immune response.

Under physiological conditions, several microbial products, such as lipopolysaccharide (LPS), bacterial DNA, and flagellin, defined as Microbe-associated Molecular Patterns (MAMPs) and Pathogen-associated Molecular Patterns (PAMPs), may be recognized by pattern recognition receptors (PRRs) expressed on dendritic cells, B cells, and T cells, as well as on enterocytes. Mucosal and mesenteric lymphoid tissues host immune effectors that respond to these gut-derived bacterial products [26,27]. The activation of several intracellular signaling pathways, including myeloid differentiation primary response 88 (MyD88), NF-kB, and nucleotide-binding oligomerization domain-containing protein 2 (NOD2) pathways, elicit the secretion of inflammatory cytokines, chemokines, vasodilatory mediators, and other regulatory factors [28,29,30,31]. Additionally, other mechanisms participate in this continuous self-balancing crosstalk in a network of bidirectional interactions shaping both the immune cell population and GM biodiversity and activity, specifically the production of short-chain fatty acids (SCFAs), bile acid metabolism, antimicrobial proteins (AMPs), and IgA secretion. SCFAs, such as butyrate, propionate, and acetate, originate from the GM fermentation of non-absorbable carbohydrates and exert well-characterized regulatory functions in the intestinal microenvironment and beyond; thus, attention has been recently focused on their therapeutic potentialities in different fields [32,33]. SCFAs play a relevant role in the modulation of both the innate and adaptive immune responses [34,35,36]; for instance, in preclinical studies, butyrate showed the ability to enhance macrophage function and to prompt the recall response of memory T cells expressing free fatty acid receptor (FFAR) 2/3 [37,38]. However, the regulatory activity of SCFAs seems more globally unbalanced toward an anti-inflammatory effect and the regulation of mucosal immunity, orchestrating the immune response through several mechanisms [39]. Among these, a considerable role is played by the inhibition of histone deacetylase by butyrate, as this interaction is able to inhibit the activation of nuclear factor kB (NF-kB) with a consequent anti-inflammatory effect [40]. This same mechanism could also be the reason for the suppressor effect of butyrate on neutrophils’ migration and the release of proinflammatory mediators and for the reduction in neutrophil extracellular traps (NETs) [41]. Butyrate and propionate also showed a tolerogenic effect after the administration of ovalbumin plus cholera toxin, associated with reduced specific antibody titers [42].

### 2.2. The Microbiota outside the Gut and Immune Response

A scenario similar to that previously described has been outlined for commensal bacteria located in other compartments; similar to the gut, the resident microbiomes of the skin, oral cavity, and respiratory tract play a significant role in immunomodulation and can affect the differentiation of T lymphocytes [43,44,45]. An emerging theme concerns the role of the lung microbiome in the immune response to respiratory diseases. The lung microbiome is less heterogeneous and consists of a smaller biomass compared to that in the gut; *Bacteroidetes* and *Firmicutes* are the dominant phyla [46]. During lung diseases, changes in the lung microbiota composition have been observed, but these modifications remain an open debate. The lung microbiota can regulate the expression of some innate immunity genes, upregulating the production of interleukin (IL)-5, IL-10, interferon (IFN) γ, and C-C motif chemokine ligand (CCL) [12]. There are also connections between the gut and the lung, which realizes the gut–lung axis. One possibility involves the translocation of bacteria via oropharynx reflux [47]; Ruane et al. reported another example after intranasal immunization with inactive cholera toxin. Lung dendritic cells induce the expression of the gut-homing molecules α4β7 and CCR9 on IgA+ B cells, a process that is mediated by retinoic acid signaling in lung IgA + B cells. These B cells migrate from the lung to the gut in response to the nasal intake of cholera toxin [24]. Furthermore, this study demonstrated that the GM influences the humoral immune response of the lung: in fact, in a germ-free mouse, the IgA response in the stool, lung, and serum following oral intake of cholera toxin is significantly reduced. However, despite the evidence of specific microbiota signatures in the bronchoalveolar fluid during viral infections [48,49,50], little data on vaccine-related changes are available. In a recent preclinical study, Bacillus Calmette–Guérin (BCG) vaccination through the subcutaneous route in mice was associated with changes in both the lung and gut microbiota in terms of alpha- and beta-diversity, as well as bacterial relative abundance [51]; of interest, to explore the behavior of the gut–lung axis, the authors conducted a multivariate analysis by comparing BCG with tuberculosis-induced changes, concluding that tuberculosis-infection-induced immune changes affected the microbiota at both the gut and lung levels, while BCG vaccination mainly modified the GM. Moreover, BCG vaccination was associated with an altered profile of microbial products capable of modulating the innate immune response in the lung via memory alveolar macrophage induction [52].

The human oral cavity harbors the second most abundant microbiota after the gastrointestinal tract, and it comprises ~700 bacterial phyla categorized into six major phyla [53]. Moreover, immune cells are present in the oral mucosa, with a large prevalence of neutrophils. Neutrophils and macrophages play a central role in oral mucosa homeostasis. Local macrophages regulate the neutrophil response by secreting IL-17 and IL-23. When neutrophil apoptosis occurs, the secretion of proinflammatory ILs is discontinued. If neutrophils are unable to move into tissues or their apoptosis is delayed (as in some pathological conditions, such as leukocyte adhesion deficiency (LAD)), IL-23 and IL-17 cytokines persist and contribute to the exacerbation of tissue inflammation [54]. Despite the large body of evidence for the involvement of the oral microbiota in the immune response to infections within the oral cavity and beyond, very poor evidence is available on its ability to modulate vaccine immunogenicity; indeed, oral hygiene interventions have been associated with better outcomes in patients with pneumonia, possibly in relation to the potential role of oral hygiene in reducing the aspiration of oral pathogens, as well as in favoring the restoration of the respiratory epithelium functioning [55,56]. Severe acute respiratory syndrome coronavirus 2 (SARS-CoV-2) has been associated with dysbiosis of the oral microbiome, which was further correlated with clinical severity, inflammatory cytokine production, and a decrease in the IgA response [57]; moreover, SARS-CoV-2 severity was inversely correlated with the abundance of *Bacteroides* in both the oral cavity and the gut [58]; in mouse models, this genus was demonstrated to reduce the expression of intestinal angiotensin-converting enzyme 2 (ACE2), a well-known receptor for SARS-CoV-2 entry into host cells [59]. Of interest, in a recent study, SARS-CoV-2 vaccination induced an alteration in the oral microbiomes of 40 healthy individuals, particularly a higher oral bacterial diversity and a significant decrease in *Bacteroides* [60], keeping the cause–effect debate still open in this context.

Other sites could also contribute to the immune response through their resident microbiota, for instance, the skin microbiota for intradermal vaccines [61], but are likely to have a limited impact compared to those mentioned above.

## 3. Vaccine Immunogenicity and Microbiota in “Edge Populations”: Infants and Elderly People

The reduced response to vaccines in two particular segments of the population constituted by infants (aged from 1 month to 1 year) and older adults (aged 65 years and older) can be partially explained by age-related changes in immunity, with a reduced immune response in these populations.

Indeed, in the case of infants and newborns, the immune system is generally more unbalanced toward the T helper (Th)-2 effector profile, with weaker Th-1 immunity and a lower production of antibodies by B lymphocytes [5]. This shift is due to multiple factors: a lower number of effector memory T cells and memory B cells [5], a lower capacity to produce multiple cytokines simultaneously after TLR stimulation, and an imbalance between IL-12 and IL-10 levels. IL-12 contributes to T-cell differentiation, and its levels are reduced in newborns and infants due to its lower production by antigen-presenting cells (APCs). IL-10 production is increased by T-suppressive regulatory cells, thus resulting in an immune-suppressive effect [5]. Therefore, infants’ immune systems are characterized by a higher immune tolerance with weaker humoral immunity and increased susceptibility to infections, particularly by intracellular microorganisms [5]. Even though this imbalance could favor bacterial colonization and prevent alloimmune reactions due to higher immune tolerance [5], it may also account for the weaker response to vaccine stimulation and the need for multiple booster doses to elicit a strong vaccine response [5].

In the first years of life, the GM undergoes complex changes according to nutritional intake during the different steps of weaning, reaching a stable composition at 2–3 years of age [62]; it is noteworthy that, in this specific population, an inadequate and unbalanced nutritional intake, exposure to antibiotics, and cesarean section may result in an altered microbial profile with consequential dysregulated responses to external disturbances [62,63,64,65]. In newborn mice, the intestinal microbiota generates a strong immune response during weaning that is fundamental for the development of the immune system; this so-called “weaning reaction”, if inhibited, can lead to pathological imprinting that drives disease susceptibility later in life [66].

It has been observed that infants with an increased abundance of gut *Bifidobacteria* have an enhanced T-cell immune response and a higher antibody serum level after vaccination, with a sustained significant response to specific vaccines at two years of age [67]. In a randomized case–control study, De Vos et al. observed that the GM composition in Ghanaian infants responding to a rotavirus vaccine was more similar to that in Dutch infants [68,69,70] than to that in Ghanaian non-responders. Furthermore, an increased abundance of *Streptococcus bovis* and a reduced abundance of *Bacteroidetes* were detected in Ghanaian responders to the rotavirus vaccine, similar to Dutch controls, compared to Ghanaian non-responders [71].

In a similar study, both Pakistan and Dutch infants who responded to a rotavirus vaccine had an increased abundance of *Proteobacteria* and *Clostridium cluster XI* in comparison with non-responder Pakistan infants [72].

Jordan et al. demonstrated that the use of *Bifidobacterium* and *Lactobacillus* in infants improves the humoral immune response [73]. A systematic review by Zimmermann and Curtis analyzed 26 interventional studies on the use of *Bifidobacterium-* and *Lactobacillus*-based prebiotics and their influence on the immune response induced by oral and parental vaccines [74], concluding that the use of these probiotics in infants has a positive effect on humoral immunity with 17 different vaccines.

Older adults and elderlies are also characterized by a weaker response to immune-stimulating agents and a higher vulnerability to infections [7,75]; the condition of “immunosenescence” is the result of several age-related modifications in both innate and adaptive immunity, with quantitative and qualitative defects in most immune cells [7]. For the innate immunity arm, there is a global defect in phagocytosis and reduced migration of neutrophils. For the adaptive immunity arm, there is reduced maturation of B cells in the bone marrow (with persistence of peripheral mature B cells), reduced differentiation of B cells into plasma cells, and an impaired response of T lymphocytes, probably linked to an oligoclonal expansion of memory T cells with a reduction in T naive cells [7]. However, in addition to these age-related dysfunctions, another hallmark of this population is the presence of a chronic, low-grade, persistent inflammatory state related to aging, which has been called “inflammaging” [7,75,76]. Immunosenescence and inflammaging have been described as two faces of the same process. Inflammaging could be the result of an accumulation of aged or senescent immune cells in peripheral tissues, which constantly produce cytokines and proinflammatory mediators, with relatively reduced functions of autophagy, proteasome activity, and other pathways able to activate inflammation [75]. Chronic inflammation plays a role in many age-related diseases, and thus, inflammaging has been proposed as one of the possible factors determining age-related disability in certain individuals [75,76]. Many variables can influence immunosenescence and inflammaging, such as genetics, previous infectious exposures, nutritional status, gender, drugs, or exercise [7,75]. Notably, gut dysbiosis has been recently identified as one of the most important of these variables. Indeed, a specific pattern of gut dysbiosis with an abundance of proinflammatory bacteria such as *Bacteroides* and *Enterobacteriaceae* and a lower production of SCFAs [7,76] is present in elderlies with a higher inflammatory burden, while these alterations are not common in centenarians, who usually harbor an anti-inflammatory GM, consisting of elements such as *Akkermansia* or *Bifidobacterium* [7,75,76]. Centenarians also present a higher GM variability compared to elderlies with higher inflammaging [75,77,78]. Interestingly, studies on centenarians from different parts of the world have shown common characteristics in GM composition in these individuals despite their different geographical birthplaces, suggesting that some bacterial species could be considered part of a “longevity signature” in the gut [69] (Table 1).

Another key point of this interplay could be represented by the continuous passage of bacteria or their fragments into the bloodstream through a leaky gut, which is called “bacterial translocation”, with a chronic antigenic stimulation that increases the number of peripheral memory T cells and leads to immune system exhaustion, contributing to immunosenescence [7,76].

Thus, gut dysbiosis may represent one of the main determinants of immunosenescence and inflammaging and, therefore, constitutes a crucial variable in the impairment of the vaccine response in elderlies; however, there is a bidirectional interaction among the GM and the immune system, and gut dysbiosis could also represent a consequence of intestinal immune dysregulation, with a possible role played by the variations in GM induced by oral vaccines [7], with other key factors played by an unbalanced diet and the use/abuse of several drugs (for example, proton-pump inhibitors, non-steroidal anti-inflammatory agents, or antibiotics) in this population [7,76].

Most vaccines are developed in a young population; it would be desirable to formulate specific vaccines for the elderly population by also intervening in the composition of the adjuvants [6].

## 4. Microbiota and Oral Vaccines

The connection between the GM composition and response to oral vaccines appears to be intuitive, but the evidence in this field has also been conflicting.

In a preclinical study, germ-free and antibiotic-treated mice exposed to rotavirus showed a lower infection rate and milder symptoms compared to wild-type mice; moreover, antibiotic administration was associated with a more stable humoral response. The reduction in the availability of microbial products following antibiotic exposure may lead to the reduced entry and presentation of rotavirus antigens, as observed for other viruses [79,80]. Interestingly, prior dextran sulfate sodium (DSS) administration was associated with a reduced duration of antibody protective levels, possibly due to chronic inflammation; according to the authors, this could give an explanation for the low efficacy of rotavirus vaccination in developing countries because of a proinflammatory microbiota composition and the continuous exposure to pathogens, causing chronic immune activation. To specifically analyze the role of the antibiotic-altered microbiome over other possible antibiotic-related effects, the immunization of 7-day-old mice born from antibiotic-treated dams resulted in the development of lower IgG-specific antibody titers compared to mice born from untreated dams; no differences were found when immunization was performed 14 days after birth [81].

The seroconversion rate of rotavirus vaccines, namely, Rotarix and RotaTeq, is very low in Ghanaian infants, as well as in Africa and Asia [82,83,84,85]. A recent observational study exploring this phenomenon in healthy infants from Ghana compared the GM of 38 rotavirus vaccine responders with that of 38 non-responders based on 4-week antibody titers after the last dose [71]; the results were further compared with a cohort of 154 Dutch infants presumed to be rotavirus vaccine responders based on clinical data demonstrating a seroconversion rate > 90% in northern European countries [70,86]. Rotavirus vaccine immunogenicity was associated with an increased abundance of *Firmicutes*, particularly bacteria related to *Streptococcus bovis*, and a decreased abundance of the *Bacteroidetes* phylum, when comparing both Ghanaian rotavirus vaccine responders with non-responders and Dutch infants with Ghanaian non-responders. The authors provided several possible explanations: *Bacteroidetes*, increased in non-responders, express a lipopolysaccharide (LPS) variant with very weak immunogenicity [87]; on the other hand, the *Streptococcus bovis–Streptococcus equinus* complex could promote a microenvironment allowing for the increased replication of the attenuated virus by providing specific antigens, as occurs for other viral infections [88,89].

A significant association between the GM composition and rotavirus vaccine immune response was also observed in Pakistan; in particular, an increase in the relative abundance of *Proteobacteria* and the *Enterobacteriaceae*-to-*Bacteroides* ratio was observed in responders [76], in line with the Ghanaian cohort. Notably, no significant results were found in other similar observational studies [90,91], although limited by a small sample size.

A recent randomized controlled trial investigated the effect of azithromycin on the immunogenicity of the serotype-3 monovalent oral poliovirus vaccine (mOPV3) in healthy Indian infants; in this study, the authors found that azithromycin administered once daily for 3 days did not improve vaccine immunogenicity, despite resulting in reduced levels of biomarkers of environmental enteropathy in developing countries and in a lower prevalence of pathogenic intestinal bacteria [92]. In China, the humoral response to OPV was associated with the abundance of *Bifidobacteria* and a lower GM diversity [93]; in the same setting in an Indian cohort, no significant association was found in the GM composition, whereas reduced diversity was confirmed in OPV responders [94].

However, conflicting data on this topic and the lack of study setting standardization clearly make these results not fully reliable and difficult to evaluate.

## 5. Microbiota and Parenteral Vaccines

While the influence of the gut microbiota on oral vaccines seems to be easier to figure out, the evidence on parenteral vaccines (i.e., vaccines administered by intradermal, subcutaneous, and intramuscular routes) is not yet well explained.

In 2019, Hagan et al. conducted a comprehensive study aiming to uncover the indirect association between the gut microbiota and vaccine immunogenicity by analyzing the immune response to influenza vaccination in young adults after a 5-day administration of broad-spectrum antibiotics [95]. The authors found a significant decrease in IgG1 and IgA responses only in adults with low basal H1N1-specific antibody titers, while no differences were found in the remaining cohort compared to controls. Additionally, in order to assess whether there was a correlation between these results and the serum metabolome, a multi-omics analysis was conducted and revealed an impaired metabolic profile in the antibiotic-treated group; bile acid and tryptophan metabolism were altered, GM diversity was lower, and a 10.000-fold reduction in the overall fecal bacterial count was also observed. The decrease in secondary bile acid production (in particular, lithocholic acid) showed a strong association with activator protein 1 (AP-1) signaling, together with other transcriptional changes accounting for an increased inflammatory response. Finally, significant bacteria–metabolite associations were found; a lower abundance of bacteria belonging to *Lachnospiraceae* and *Ruminococcaceae* families after antibiotic treatment was associated with impaired bile acid metabolism and IgG1 response to H1N1.

Concerns about the influence of the GM on parenteral vaccines could be at least partially ascribed to the fact that the populations studied are represented by small sample sizes of adults, while vaccinations usually concern infants and the elderly; in addition, these studies are commonly based on short-term antibiotic-driven perturbations of the microbiota. However, considering that 50% of infants were found to be exposed to 5-day antibiotic therapy or more during their first year of life in a Western cohort [96], the analysis of GM dysbiosis in this setting is more consistent with real-life scenarios. Indeed, in a recent preclinical study, early-life antibiotic exposure in mice resulted in sustained changes in the GM composition and subsequently impaired antibody responses to five human vaccines [97]; in the treated group, the GM composition was characterized by a reduced abundance of *Bacteroidetes* and subsequent colonization by *Lachnospiraceae*, *Enterobacteriaceae*, and *Akkermansia* spp. The authors also showed that the immune response to vaccines was similar to that of untreated mice, both chronic-antibiotic-exposed mice and antibiotic-exposed mice, after fecal microbiota transplantation (FMT) from untreated mice donors, further confirming the relevance of the recolonizing GM rather than that of the antibiotic exposure per se in the modulation of the humoral response to vaccines. Similar data were obtained in models of non-antibiotic-driven dysbiosis; for instance, TLR5 double-knock-out mice showed a reduction in both antibody titers and specific plasma cell differentiation after the influenza vaccine [98]. The administration of flagellated *Escherichia coli* to germ-free or antibiotic-exposed mice restored a normal antibody response, as opposed to the administration of a-flagellated *Escherichia coli* [98]. Notably, in the same study setting, a similar function of the flagellin/TLR5 interaction was observed with another subunit vaccine, the inactivated polio vaccine.

In line with these results, in a prospective observational study conducted on 249 infants from Bangladesh, intestinal *Bifidobacteria* abundance was significantly associated with CD4+ T-cell responses and antibody responses to several parenteral vaccines at 2 years of age, confirming that gut microbial ecology may also modulate human immune memory cells to parenteral vaccines [99].

In summary, studies trying to dissect the role of the GM in the response to parenteral vaccines are often based on antibiotic-related interventions and lack any focus on molecular mechanisms or on other tools for microbiota modulation, allowing only indirect and speculative conclusions.

## 6. Pathophysiological Mechanisms of the Influence of the Microbiota on Vaccine Response

The potential role of the microbiome in the modulation of vaccine immunogenicity takes root in a number of findings from the past two decades, gradually disclosing several models of microbial modulation of the immune response, first in general and then specifically in vaccines. In 2008, Hall et al. set a milestone in the link between the microbiota and immunity by conducting an elegant study showing, for the first time, how the DNA of commensal intestinal bacteria acts as a natural vaccine adjuvant, promoting the production of effector T cells through the activation of TLR9 in animal models in both in vitro and in vivo experiments. Conversely, in this study, TLR9-deficient mice showed an increase in regulatory T cells, with a consequent impairment in the immune response to both oral Microsporidia infection and vaccination with the mutant form of *Escherichia coli* labile toxin (LT) as a mucosal adjuvant; additionally, antibiotic administration in TLR9−/− mice resulted in a worse immune response to oral infection [100]

Several observational studies later showed the association between specific bacterial strains and vaccine efficacy [67,71,75], outlining different models of microbial immunomodulation, including the presentation of cross-reacting epitopes and the production of microbial products enhancing T-cell activity, promoting a sustained immune response [70,98,101,102,103]. It is well established that, taking into account the totality of human antigens, MAMPs, and PAMPs, there is a significant overlap in the tridimensional characteristics of the epitopes presented to T-cell receptors (TCRs) that accounts for T-cell cross-reactivity [103]. In fact, a specific TCR binding site can cross-recognize different microbial and self-antigens, thereby generating clones of memory T cells that are able to react to previously unrecognized microbial antigens [104,105,106,107]. In a group of healthy volunteers seronegative for the challenge influenza viruses H3N2 or H1N1, memory CD4+ T cells responded to viral peptides from the core proteins of these challenge strains and were related to a better disease course [108]. Similarly, individuals who were never exposed to human immunodeficiency virus (HIV-1), cytomegalovirus (CMV), and herpes simplex virus (HSV) had memory cells that reacted to peptide-major histocompatibility complex (pMHC) tetramers derived from the specific viral antigens [107]. The GM can sustain a persistent pool of cross-reactive T and B lymphocytes and thereby influence immune responses to infections [99,109]. In a recent study, it was observed that memory T cells derived from the presentation of a protein of a prophage in the genome of the bacteriophage *Enterococcus hirae* can cross-react with tumor-associated antigens and stimulate improved responses to cancer immunotherapy in humans [110]. Based on this evidence, it is possible to speculate that microbiome-derived epitopes can simulate the antigenic determinant of a specific vaccine and, consequently, enhance its immunogenicity.

Finally, the GM acts as a provider of “local” vaccine adjuvants with an ancillary function in the enhancement of adaptive immunity, thereby promoting a stronger and enduring response. Vaccine adjuvants are chemical molecules that generally act as PRR ligands, promoting the increased activation of PRR-expressing cells, such as macrophages and dendritic cells, and improving vaccine immunogenicity. The most widely used vaccine adjuvants are TLR3 and cytosolic ribonucleic acid helicase (RLR) ligands, such as synthetic analogs of dsRNA (i.e., Poly IC); TLR4 ligands, such as LPS and the detoxified derivative monophosphoryl lipid A (MPL); TLR5 ligands (i.e., bacterial flagellin); TLR7 and TLR8 ligands (i.e., guanosine- and uridine-rich ssRNA); and TLR9-synthetic 18–25-base oligodeoxynucleotides (ODNs) with optimized CpG motifs (CpG-ODN) [111]. It has been shown in animal models that the interaction between microbial antigens and specific TLR isoforms is essential to building an optimal antibody response to non-adjuvanted vaccines, but the data are conflicting [87,98]. Among adjuvants, the role of LPS is certainly the most well characterized in the literature; LPS variants are associated with a different range of stimulation effects on the innate immune response through TLR4 sensing, thus giving a rationale for the different immunogenicity of specific commensal microbial communities [87]. From this view, LPS can represent an adjuvant in the vaccine immunization process [112]. Other microbial products activating PRR pathways have been shown to be needed for an adequate immune response; for example, the immune response to intranasal immunization with cholera toxin was impaired in germ-free or antibiotic-treated mice, while reconstitution with a NOD2 agonist or *Staphylococcus sciuri* was sufficient to elicit proper dendritic cell activation via the recognition of peptidoglycan molecules containing muramyl dipeptide (MDP). As previously described [98], in humans, the early expression of TLR5 correlates with antibody titers in response to the trivalent inactivated influenza vaccine (TIV), suggesting a possible role of flagellin in adjuvating the vaccine response in this specific setting [113]. TLR5 specifically recognizes flagellin, another microbial component expressed by flagellated Gram-negative bacteria, further confirming the role of this microbiota-dependent pathway in vaccines with no or weak adjuvants.

## 7. Promising Tools for the Modulation of the Microbiota for the Enhancement of Vaccine Immunogenicity

The strong link of the GM with the immune response and, therefore, with vaccine immunogenicity has stimulated numerous speculations on what role GM modulation might have in this scenario [96,114]. Although the role in clinical practice still appears unclear, below, we discuss the main modulators of the GM that could be used in eliciting an effective vaccine response.

### 7.1. Prebiotics, Probiotics, and Synbiotics

Prebiotics (i.e., dietary fibers and fermented foods) are substrates utilized by host microorganisms that confer health benefits. They are generally represented by short-chain carbohydrates used as substrates for the growth of beneficial bacteria in the upper gastrointestinal tract, but they also include non-carbohydrate substances (i.e., cocoa-derived flavonols). Probiotics are live microorganisms able to transfer a health advantage to the host [115].

A recent systematic review evaluated 26 randomized, placebo-controlled clinical trials on the effect of probiotic administration on the vaccine response, showing discordant results [116]. Indeed, only 3 studies out of 7 demonstrated a benefit in the response to parental vaccination in infants and children, while a better response to the influenza vaccine was shown in 5 studies out of 12.

In a single study conducted on 123 healthy adults receiving the hepatitis A virus vaccine, a significant improvement in the vaccinal response was observed with the administration of probiotics [117], while another study conducted on pregnant women receiving probiotic supplements showed no benefits in the vaccine responses of their infants, even suggesting a potentially negative effect [118].

However, besides the heterogeneity of the populations, these discordant results may be due to the fact that most of the selected studies had heterogeneous settings, involving a total of 40 probiotics (both live and heat-killed) administered before 17 different vaccines across all age groups. Notably, a beneficial effect of probiotics was suggested with oral and influenza vaccines in elderly people. In other studies exploring the effect of *Lactobacillus* spp. on influenza vaccination in elderlies, a beneficial effect was observed mainly in terms of the humoral response, still without exploring the actual protective role against the influenza virus [74,119,120]. In a small monocentric study conducted on 42 elderlies who were randomized to receive either a *Lactobacillus paracasei*-based dairy product or a placebo, significantly higher antibody titers after receiving influenza vaccination were observed in a subgroup of the oldest patients (>85 years) receiving the probiotic-based product [119]. Similar results were confirmed in a larger randomized, multi-center trial conducted on a total of 241 elderly adults from different nursing homes who were randomized to receive either a probiotic fermented dairy drink containing a *Lactobacillus casei* strain or a non-fermented dairy product, in which the group consuming the probiotic-based product also showed a significantly higher antibody titer after influenza vaccination, even if only for the B influenza strain [120]. However, these data are confounding and difficult to evaluate, considering the absence of *Lactobacillus* spp. in the microbiota of healthy adult individuals and their potentially double-sided effects on the gut microbial ecology [121].

Synbiotics (both nutraceutical and bacterial products made of a mixture of pre- and probiotics) are also gaining attention in this field. In a recent preclinical study, supplementation with spirulina, amaranth, flaxseed, and micronutrients in mice colonized with the microbiota of children from Bangladesh increased the mucosal IgA response to an oral cholera vaccine [122]; moreover, hyporesponsive mice co-housed with treated mice also developed a better response to vaccination; the authors isolated five of the transferred bacterial species between the co-housed groups (*Bacteroides acidifaciens*, *Bacteroides fragilis*, a non-toxigenic *Clostridioides difficile*, *Clostridium innocuum*, and *Fusobacterium mortiferum*) and administered them as a bacterial consortium together with the nutraceutical product, obtaining a further improved response in recipient mice and confirming the main role of the microbiota as the conductor of immune modulation.

On the other hand, even though the use of probiotics, prebiotics, and synbiotics has been considered generally safe in most healthy individuals, some studies have highlighted potential risks in their use in specific populations [123]. In young infants, the administration of probiotics has been associated with a higher risk of mucosal infections, as well as with an increased percentage of colonization in newborns by vancomycin-resistant *Enterococcus* [124,125]. Furthermore, in critically ill or immunocompromised patients, probiotic administration was associated with a higher infection risk and mortality, mainly due to sepsis or fungemia [126,127].

However, it is noteworthy that most of the largest clinical trials concerning the use of these products were designed to evaluate the clinical efficacy rather than the safety, and many of them did not adequately report the exact rate of probiotic- or prebiotic-related adverse events; in several studies, they were not reported at all [128,129]. Thus, while randomized trials designed to evaluate the safety of the wide use of prebiotics or probiotics are lacking, a critical issue could be represented by their use in specific subgroups of patients, such as young infants, critically ill patients, or frail individuals.

### 7.2. Diet

A recent study analyzed the possible role of diet modulation in modifying the GM composition and the vaccine response; in gluten-free-diet mice, the IgG response to the first dose of the tetanus vaccine was reduced in comparison with controls, in association with an increased relative abundance of *Bifidobacteria*; moreover, both IgG levels and *Bifidobacteria* abundance in the gluten-free mice showed broad variability. However, after a booster vaccination, no statistical difference in the IgG response was confirmed between the groups [130].

Notably, the authors outlined how the gluten-free diet was associated with a higher number of Treg cells, consistent with previous evidence [131,132], thereby providing a possible explanation for the restrained immune response.

In another study, a diet based on oligosaccharides also resulted in a reduced IgG response in a similar setting [133], following its capability to increase the expression of well-known anti-inflammatory microbial strains, including *Bifidobacteria*, *Lactobacilli*, and *Akkermansia* [134,135,136].

## 8. Future Perspectives

The increasing comprehension of the interplay between the microbiota and the immune response could represent an opportunity to enhance vaccine immunogenicity.

The use of microbial strains as vectors for antigens is an emerging tool in the field of vaccine bioengineering. For instance, in 2022, Zhang et al. conducted a study to explore the characteristics of the immune response resulting from the oral administration of a recombinant yeast (*Saccharomyces cerevisiae*) exhibiting the SARS-CoV-2 spike protein, used as both a vector and a biological adjuvant. The recombinant yeast elicited a specific mucosal and systemic response and stimulated a peculiar gut microbiota composition when compared with the gut microbiota in mice receiving wild-type yeast cells. A remarkably higher abundance of bacteria belonging to *Firmicutes*, *Verrucomicrobia*, *Bacteroidetes*, and *Actinobacteria* was described in mice treated with the recombinant yeast than in the control group. Furthermore, sex differences were observed, with a prevalence of *Succinivibronaceae*, *Akkermansiaceae*, and *Atopobiaceae* in the gut microbiota of male mice and *Desulfovibrionaceae* in females [137]. As this study suggests, bioengineering approaches may improve immune responses by acting on the gut microbiota. In the field of vaccine immunogenicity, administering safe bioengineered products could be a valuable resource, first with the preclinical use of engineered tissues or organoids in order to predict microbial changes and vaccine responses [138].

Small extracellular vesicles derived from milk have been recently studied as carriers of pharmaceuticals. Since they seem to play a favorable role in human health as immune regulators and antioxidants [139], a biomedical evaluation of their applications in vaccination strategies could be interesting. Indeed, human administration of vaccines carried by milk-derived vesicles may provide an improved gut microbiota composition and immune regulation.

In addition, in recent years, “superfoods” have gained growing attention as foods capable of fortifying the immune system due to their richness in advantageous nutritional components. Some fruits and vegetables, nuts, spices, salmon, and honey, among others, have been defined as superfoods [140,141]. However, microalgae like chlorella and spirulina have shown valid nutritional value compared with common foods. They are highly rich in proteins, vitamins, iron, essential fatty acids, and polysaccharides [142]. In the fight against COVID-19, fortified foods with additional micronutrients, encapsulated foods (i.e., milk with soya-lecithin-encapsulated calcitriol), and vegetal foods have been useful for improving antiviral immunity [143]. Therefore, more research and, hopefully, clinical trials need to clarify the potential role of superfoods in enhancing vaccine immunogenicity.

Improving infant formula with specific supplements could be another topic for further investigation. Alliet et al. studied the effects of infant formula with the probiotic *Limosilactobacillus reuteri* and the addition of fucosyllactose. They described significant changes in the gut microbiota, with increased beneficial *Bifidobacteria* and reduced harmful *Clostridioides*, similar to what happens after breastfeeding [144]. Moreover, Estorninos et al. studied the impact of the addition of oligosaccharides from bovine milk to infant formula on the intestinal microbiota. Their results showed an overgrowth of *Bifidobacteria*, a reduction in pathogenic microbes, and the general amelioration of intestinal immune regulation. In particular, IgAs induced by the polio vaccine were significantly higher in the group receiving the supplemented formula compared with the control group [145]. In the next few years, the use of enhanced formula in infant feeding could increase vaccine immunogenicity through favorable gut microbiota changes.

On the other hand, limited data are available regarding the potential efficacy of FMT on vaccine immunogenicity. A study conducted on broiler birds suggested that gut microbiota is responsible for the effectiveness of the *Escherichia coli* vaccine expressing the *Campylobacter jejuni* N-glycan. In particular, the presence of species belonging to *Clostridium* and *Ruminococcaceae* leads to a better response. The authors also showed that birds transplanted with the gut microbiota of successfully vaccinated birds developed better vaccine-related immunoglobulin responses when compared to non-transplanted birds [146]. In humans, FMT might be a disproportionate tool in this context, not cost-effective, widely unavailable, and less practical compared to the other strategies analyzed above. Its role could be considered in specific settings, particularly in developing countries, where it could be more beneficial. As described by Srivastava and colleagues, the reduced effectiveness of the rotavirus vaccine in poorer countries can be in part attributed to the altered microbiota composition. They found increased *Enterococcus* and *Proteus* and reduced *Bifidobacterium*, *Streptococcus*, and *Clostridium* in their pig model transplanted with an undernourished human microbiota [147]. As previously reported, the immunomodulatory properties of the gut microbiota influence vaccine responses, resulting in higher immunogenicity in healthy conditions and impaired immune regulation in conditions of dysbiosis. In this situation, FMT could reasonably represent a useful tool for enhancing vaccine responses by restoring the physiological gut microbiota with a higher microbial diversity, an increased number of beneficial species, and a reduction in pathogenic bacteria. However, FMT requires an accurate donor selection and microbiological tests in order to avoid the transmission of infectious diseases [148]. In developing countries, where fecal microbiota transplants are carried out less frequently, achieving safe procedures could be even more difficult. In addition, the use of FMT for enhancing vaccine immunogenicity, which means preventing infections, rather than focusing on its more significant application in the treatment of diseases (e.g., recurrent *Clostridioides difficile*-associated colitis), could be another matter of debate.

In conclusion, vaccination efficacy is influenced by the intrinsic characteristics of the vaccine, the host immune system, and environmental elements; in this interplay, the modulation of the gut microbiota represents a promising path to make a real difference; however, it is necessary to explore the use of specific bacterial strains (i.e., *Bifidobacteria* and *Lactobacilli*) as natural adjuvants [149] and to pursue the identification of other bacterial species that influence the response to vaccination through DNA extraction and next-generation sequencing techniques [150,151]. Moreover, further investigations are needed to identify, with the help of next-generation sequencing issues, specific microbiota compositions that better enhance specific immune responses to specific vaccines, built by using new advanced immunoinformatic methodologies, in order to rapidly design a more personalized, safe, and effective type of vaccine that can elicit a strong and long-lasting immune response, especially in fragile people.

Overall, further randomized controlled studies are needed to evaluate the clinical effects of these interventions, particularly those targeting large cohorts of populations with dysbiosis (infants in developing countries and elderlies) with standardized administration schemes and a proper analysis of the microbial ecology (-omics statistical analysis), also extending it to microorganisms other than bacteria that are likely to have a relevant impact on vaccine immunogenicity [152].

## Figures and Tables

**Figure 1 vaccines-11-01609-f001:**
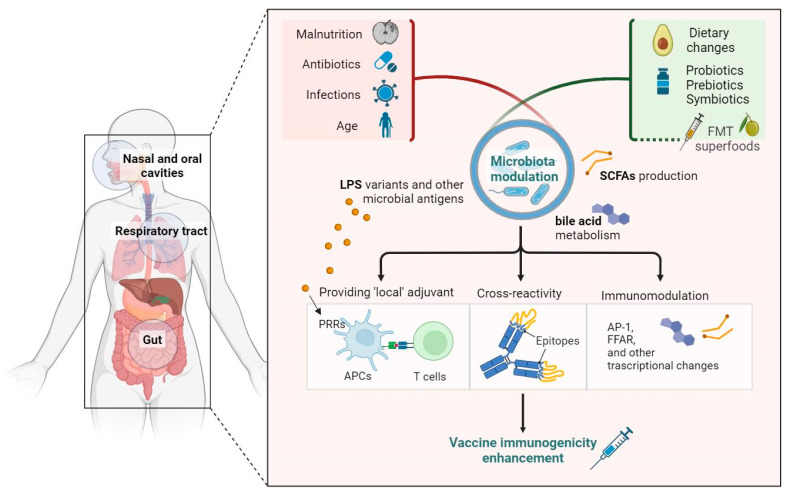
Several factors participate in lowering the efficacy of vaccination, with growing evidence supporting the role of the microbiota as a mediator of this interplay. Targeting the resident microbiota of nasal and oral cavities, respiratory tract, and gut could therefore be the key to unlocking the immune response in specific subpopulations. Abbreviations: AP-1, activator protein 1; FFAR, free fatty acid receptors; FMT, fecal microbiota transplantation; LPS, lipopolysaccharides; PRR, pattern recognition receptors; SCFAs, short-chain fatty acids.

**Table 1 vaccines-11-01609-t001:** Microbial signatures in GM in healthy older adults and centenarians. SCFAs, short-chain fatty acids.

Gut Microbiota Characteristics and Composition	Centenarians
Microbial diversity	Increased [75]
Production of SCFAs	Increased [75,77]
Bacterial capability for glycolysis	Increased [75,77]
*Akkermansia* spp.	Increased [75]
*Bifidobacterium* spp.	Increased [75]
*Christensenellaceae* spp.	Increased [75]
*Synergistaceae* family	Increased [77]
*Ruminococcaceae* family	Decreased [78]
*Lachnospiraceae* family	Decreased [78]
*Bacteroidaceae* family	Decreased [78]

## Data Availability

Not applicable.

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
