# Peer review of "Factors Influencing Microbiota in Modulating Vaccine Immune Response: A Long Way to Go"

_vaccines, 2023, doi:10.3390/vaccines11101609_

Round 1

Reviewer 1 Report

The manuscript by the Cianci group summarizes very well most of the factors influencing the microbiota in the modulation of the immune response to vaccines. This is a novel an interesting issue that has to be considered together with additional factors in the design of vaccines. I think this manuscript should be published in the journal. 

Perhaps, a few general guidelines that may be considered regarding the microbiota and the vaccine design should be included in the text.

Author Response

Rome, 29th September 2023

Dear Editor of “Vaccines”,

first of all, my coauthors and I would like to thank You sincerely for this opportunity of cooperation, following the submission of the paper “Factors influencing microbiota in modulating vaccine immune response: a long way to go” and its possible publication upon “Vaccines”.

We profoundly thank the reviewers for the comments and useful suggestions aimed at improving the final version of the paper.

This is a point-by-point list of changes made in the paper:

REVIEWER 1

A few general guidelines that may be considered regarding the microbiota and the vaccine design should be included in the text.

Thanks for your suggestion. We added some considerations about vaccine design from a microbial perspective in the section 8 of the paper (line 644-655).

We thank You for your constructive critique and we hope the review process has led to an improved manuscript.

If additional changes are warranted, we will make them.

We hope that this revised version of our manuscript may now be found suitable for publication.

Sincerely,

Rossella Cianci, MD, PhD

Reviewer 2 Report

 The manuscript should be revised and following points should be improved. 

Abstract:

Lack of Clear Objectives: The abstract should explicitly state the objectives of the study. It mentions exploring the interaction between microbiota and the immune system but doesn't specify the primary research question or hypothesis.

Introduction:

The introduction is quite lengthy and includes substantial background information. While background is essential, it could be more concise to maintain reader engagement.

The introduction relies heavily on citations, making it dense and challenging to read. Some information could be summarized without citing specific studies.

The transition from the general discussion of vaccination to the role of microbiota could be smoother. It's important to establish the context more clearly.

There is inconsistency in citation style (e.g., some citations in square brackets, some in parentheses). A consistent citation style should be followed.

Microbiota and immune modulation:

The section delves into technical details about the microbiota and immune system interaction without providing a clear connection to the broader context of vaccine immunogenicity.

The section uses complex terminology and may not be accessible to readers without a strong background in microbiology or immunology.

Vaccines Immunogenicity and Microbiota in 'edge populations':

The section title "edge populations" is not immediately clear. It should be defined or explained within the section.

The section makes claims about the influence of microbiota on vaccine response in infants and elderly people but lacks specific references to studies supporting these claims.

Microbiota and oral vaccines:

The text includes incomplete sentences (e.g., "possibly through the aspiration of oral pathogens, or the restoration of the respiratory epithelium functioning"). These should be revised for clarity.

The section mentions conflicting evidence but does not provide specific examples or studies to illustrate these conflicts.

Microbiota and parenteral vaccines:

The section lacks clear explanations about what "parenteral vaccines" are, assuming prior knowledge from the reader.

The section includes multiple abbreviations (e.g., GM, FMT) without prior explanation or definition.

While the section mentions studies related to antibiotic-driven dysbiosis and vaccine response, it doesn't cite specific studies or provide details on their findings.

Pathophysiological mechanisms of the influence of the microbiota on vaccine response

The section lacks clarity and structure, making it difficult to follow the flow of the argument. E.g., the section should start with a clear introduction and purpose, followed by a structured presentation of key concepts.

Some claims are made without proper citations, which can weaken the credibility of the argument. E.g., Claims such as "Several observational studies have shown the association between specific bacterial strains and vaccine efficacy" should be supported by specific references.

The manuscript uses many abbreviations (e.g., GM, MAMPs, PAMPs, TLRs) without consistently defining them, which can be confusing for readers.

Promising tools of modulation of the microbiota for the enhancement of vaccine immunogenicity

The section discusses various tools for modulating the microbiota but does not address potential limitations or challenges associated with these approaches. E.g., It would be beneficial to discuss potential drawbacks or side effects of prebiotics, probiotics, and synbiotics.

While mentioning the effects of prebiotics and probiotics on vaccine response, the section lacks in-depth discussion of clinical evidence, including study design, sample sizes, and statistical significance. E.g., Instead of simply stating that "a beneficial effect was observed," provide specific data and references to clinical trials.

Future perspectives

The section mentions various emerging tools and strategies but doesn't clearly state their implications for vaccine development or clinical practice. E.g.,  It would be helpful to explicitly mention how these future perspectives might change vaccine development or public health strategies.

The discussion of fecal microbiota transplantation (FMT) is somewhat vague and could benefit from more in-depth analysis of its potential role in enhancing vaccine responses. e.g., Discuss the practicality, safety, and ethical considerations associated with using FMT for vaccine enhancement.

Overuse of Abbreviations: The section continues to use a high number of abbreviations without consistent explanations.

Author Response

Rome, 29th September 2023

Dear Editor of “Vaccines”,

first of all, my coauthors and I would like to thank You sincerely for this opportunity of cooperation, following the submission of the paper “Factors influencing microbiota in modulating vaccine immune response: a long way to go” and its possible publication upon “Vaccines”.

We profoundly thank the reviewers for the comments and useful suggestions aimed at improving the final version of the paper.

This is a point-by-point list of changes made in the paper:

REVIEWER 2

  • Abstract:

Lack of Clear Objectives: The abstract should explicitly state the objectives of the study. It mentions exploring the interaction between microbiota and the immune system but doesn't specify the primary research question or hypothesis.

Thank you for your suggestion. In addition to what has already been stated, this review is meant to display all the current evidence on the immune-modulating role of the gut microbiota in the specific setting of vaccination, with the aim of laying the foundations for the development of more robust future data on the modulation of the human microbiota as a tool to enhance vaccine response. We agree with your opinion and modified this part to be more accurate.

  • Introduction:

The introduction is quite lengthy and includes substantial background information. While background is essential, it could be more concise to maintain reader engagement.

Thank you for your observation. We provided a shorter and more concise version of the introduction.

The introduction relies heavily on citations, making it dense and challenging to read. Some information could be summarized without citing specific studies.

We synthesized part of the information provided and reduced the number of papers cited.

The transition from the general discussion of vaccination to the role of microbiota could be smoother. It’s important to establish the context more clearly.

Thank you for your suggestion. We tried to make more comprehensible the transition from the background to the role of microbiota.

There is inconsistency in citation style (e.g., some citations in square brackets, some in parentheses). A consistent citation style should be followed.

We checked the citation style in order to correct the mistakes.

  • Microbiota and immune modulation:

The section delves into technical details about the microbiota and immune system interaction without providing a clear connection to the broader context of vaccine immunogenicity.

The connection between microbiota, immune system, and vaccine immunogenicity has been discussed in the following paragraphs to better elucidate these topics.

The section uses complex terminology and may not be accessible to readers without a strong background in microbiology or immunology.

The terminology has been made more accessible and the acronyms have been written explicitly.

  • Vaccines Immunogenicity and Microbiota in 'edge populations':

The section title "edge populations" is not immediately clear. It should be defined or explained within the section.

The section makes claims about the influence of microbiota on vaccine response in infants and elderly people but lacks specific references to studies supporting these claims.

Thanks to the Reviewer for allowing us to clarify this point. The concept of edge population was clarified by explaining the age groups.

Specific references have been included regarding the influence of the microbiome in the vaccination of the two age groups.

  • Microbiota and oral vaccines:

The text includes incomplete sentences (e.g., "possibly through the aspiration of oral pathogens, or the restoration of the respiratory epithelium functioning"). These should be revised for clarity.

We fixed the meaning of the sentence which was indeed unclear and reviewed the 4th paragraph to verify that there is no incomplete sentence.

The section mentions conflicting evidence but does not provide specific examples or studies to illustrate these conflicts.

Thanks for giving us the opportunity to clarify. The aim of this section is to show the consistent and robust evidence available on the subject, with some evidence in favor and some against a significant positive effect of microbiota composition/modulation on vaccine efficacy.

Among the others cited, the ones that are displayed in this section are the following:

- suggesting a relevant role of microbiota on vaccine immunogenicity 71, 70, 93.

- against a relevant role of microbiota on vaccine immunogenicity 92, 94.

  • Microbiota and parenteral vaccines:

The section lacks clear explanations about what "parenteral vaccines" are, assuming prior knowledge from the reader.

Thanks for your advice, we strongly agree. We gave a definition of parenteral vaccine in the 2nd row of the 5th paragraph (line 360).

The section includes multiple abbreviations (e.g., GM, FMT) without prior explanation or definition.

We have verified that all the abbreviations in this section were previously clarified in the text.

While the section mentions studies related to antibiotic-driven dysbiosis and vaccine response, it doesn't cite specific studies or provide details on their findings.

Thanks for your suggestion. We completely agree, as there is no consistent data linking the microbiota and the parenteral vaccines beyond the ones analyzing antibiotic-driven dysbiosis. For this reason, we made it more explicit in the last sentence of the paragraph (line 408).

  • Pathophysiological mechanisms of the influence of the microbiota on vaccine response:

The section lacks clarity and structure, making it difficult to follow the flow of the argument. E.g., the section should start with a clear introduction and purpose, followed by a structured presentation of key concepts.

Thanks for your suggestion which is well appreciated, we completely agree. We made a more consistent introduction to the paragraph, whose purpose is to explain the story behind the molecular mechanisms linking the microbiota and the immune response and vaccines. We also made the text more fluent to read.

Some claims are made without proper citations, which can weaken the credibility of the argument. E.g., Claims such as "Several observational studies have shown the association between specific bacterial strains and vaccine efficacy" should be supported by specific references.

This sentence refers to some studies that are already cited before in the text and fully explained, thus it would be redundant to explicit them again. The argument is later strengthened with the subsequent studies further exploring this topic.

The manuscript uses many abbreviations (e.g., GM, MAMPs, PAMPs, TLRs) without consistently defining them, which can be confusing for readers.

We verified that all the abbreviations in this section were previously clarified in the text.

Moreover, MAMPs, PAMPs, and TLRs are widely used abbreviations in articles regarding the gut microbiota and the immune response; on the other hand, GM (gut microbiota) was necessarily used as this is used many times in the text, and it was used as an abbreviation for gut microbiota as well in other papers.

  • Promising tools of modulation of the microbiota for the enhancement of vaccine immunogenicity

The section discusses various tools for modulating the microbiota but does not address potential limitations or challenges associated with these approaches. E.g. It would be beneficial to discuss potential drawbacks or side effects of prebiotics, probiotics, and synbiotics.

Thank you for this suggestion. We added a brief discussion regarding the potential side effects of these products at the end of the paragraph (line 536).

While mentioning the effects of prebiotics and probiotics on vaccine response, the section lacks in-depth discussion of clinical evidence, including study design, sample sizes, and statistical significance. E.g., Instead of simply stating that “a beneficial effect was observed”; provide specific data and references to clinical trials.

We added more detailed information regarding some of the cited papers about this issue.

  • Future perspectives:

The section mentions various emerging tools and strategies but doesn't clearly state their implications for vaccine development or clinical practice. E.g.,  It would be helpful to explicitly mention how these future perspectives might change vaccine development or public health strategies.

Thanks for your suggestion. We made a clearer reflection on the potential clinical applications of these interventions, explaining how they might change vaccination strategies in the next few years (line 629).

The discussion of fecal microbiota transplantation (FMT) is somewhat vague and could benefit from more in-depth analysis of its potential role in enhancing vaccine responses. e.g., Discuss the practicality, safety, and ethical considerations associated with using FMT for vaccine enhancement.

Thanks for your advice. The aim of this section is focusing on several emerging tools, whose descriptions in the scientific literature are still marginal, with the hope that they can be topics for future investigations. Therefore, we are still far from a complete comprehension of their effective impact on human health, in particular in the field of vaccination. However, we agree with you that fecal microbiota transplantation involves several issues, so we made a more critical discussion about this strategy in the context of vaccine enhancement (line 629).

Overuse of Abbreviations: The section continues to use a high number of abbreviations without consistent explanations.

All the abbreviations used have been clarified in the text.

We thank You for your constructive critique and we hope the review process has led to an improved manuscript.

If additional changes are warranted, we will make them.

We hope that this revised version of our manuscript may now be found suitable for publication.

Sincerely,

Rossella Cianci, MD, PhD

Reviewer 3 Report

The review entitled “Factors influencing microbiota in modulating vaccine immune response: a long way to go” is a very interesting review and well written (maybe one or two sentences to be re-written). While the review is quite comprehensive, I still believe that the authors are leaving one significant factor that is also very well known to modulate the response to vaccine potentially via modulating microbiota: mental health (stress/anxiety/depression) of the person. Indeed, stress and emotions can affect the secretion of gastric acid, bile and mucus which then will affect the type of bacteria which can strive. Maybe a short paragraph could be added to discuss this.

More specifically could the authors address the following:

-       Line 107, it is not clear whether butyrate plays a positive role or a negative in modulating the immune response as two opposite arguments are given here. What is the opinion of the authors as there must be more than two studies on these types of SCFAs

-       Line 121 The sentence starting with “During lung diseases, it is observed a shift..” needs to be re-written.

-       Line 128, can the authors name the gut-homing molecules expressed on IgA+ B cells? Also, could the authors speculate on how the migration of lung dendritic cells to the gut provides protection against oral challenge with cholera toxin? What is the microbiota signatures in the bronchoalveolar fluid during viral infections?

-       Line 149 a bracket is missing after (LAD)

-       On few occasions (for example line 203) the authors mentioned how responses against some viruses were associated with an increased abundance of a certain of a certain type of bacteria. Can the authors clarify whether the increase was observed after the vaccination in people who responded or whether people who responded has an increase abundance in these bacteria compared to those who did not respond before the vaccination.

-       Line 237 the authors mention “common characteristics in GM composition” for centenarians. It might be good to provide a table of such a composition.

-       Line 374 is it not possible that these healthy volunteers seronegative for a specific virus were seropositive for a different virus which shared enough similarities in their protein composition (or peptides structure derived from the virus) to have T-cells which cross-react with another virus?

-       Line 396 the authors might want to add TLR9 to the text.

Very well written. Just one sentence to be re-written and just another read through the entire review.

Author Response

Rome, 29th September 2023

Dear Editor of “Vaccines”,

first of all, my coauthors and I would like to thank You sincerely for this opportunity of cooperation, following the submission of the paper “Factors influencing microbiota in modulating vaccine immune response: a long way to go” and its possible publication upon “Vaccines”.

We profoundly thank the reviewers for the comments and useful suggestions aimed at improving the final version of the paper.

This is a point-by-point list of changes made in the paper:

REVIEWER 3

The review entitled “Factors influencing microbiota in modulating vaccine immune response: a long way to go” is a very interesting review and well written (maybe one or two sentences to be re-written). While the review is quite comprehensive, I still believe that the authors are leaving one significant factor that is also very well known to modulate the response to vaccine potentially via modulating microbiota: mental health (stress/anxiety/depression) of the person. Indeed, stress and emotions can affect the secretion of gastric acid, bile and mucus which then will affect the type of bacteria which can strive. Maybe a short paragraph could be added to discuss this.

Thank you for your suggestion.

The link between mental health and microbiota is undoubtedly one of the most fascinating fields of investigation and has an established role in influencing microbiota composition.

However, our review is focused on the evidence specifically regarding microbiota and vaccines; consequently, as there is no study directly or marginally involving mental health in the interplay between microbiota and vaccines, we thought it could be possibly beyond the purposes of this review.

Anyway, we strongly agree that it is worth mentioning, and we added this topic to enrich the introduction. (line 57)

More specifically could the authors address the following:

-       Line 107, it is not clear whether butyrate plays a positive role or a negative in modulating the immune response as two opposite arguments are given here. What is the opinion of the authors as there must be more than two studies on these types of SCFAs

We fixed this part to provide a more comprehensive opinion on SCFAs.

-       Line 121 The sentence starting with “During lung diseases, it is observed a shift..” needs to be re-written.

As suggested, we rewrote the sentence.

-       Line 128, can the authors name the gut-homing molecules expressed on IgA+ B cells? Also, could the authors speculate on how the migration of lung dendritic cells to the gut provides protection against oral challenge with cholera toxin? What is the microbiota signatures in the bronchoalveolar fluid during viral infections?

Thanks for giving us the opportunity to further specify these details. We have listed which molecules are expressed on IgA+ B cells (line 161) and described the dialogue between pulmonary and intestinal immunity following oral intake of cholera toxin.

-       Line 149 a bracket is missing after (LAD)

We added a bracket.

-       On few occasions (for example line 203) the authors mentioned how responses against some viruses were associated with an increased abundance of a certain type of bacteria. Can the authors clarify whether the increase was observed after the vaccination in people who responded or whether people who responded has an increase abundance in these bacteria compared to those who did not respond before the vaccination.

Thanks for your advice. We have rewritten the ambiguous sentences at the lines 238, since the increased abundance of these bacteria is a condition that precedes the vaccination and leads to higher vaccine responses.

-       Line 237 the authors mention “common characteristics in GM composition” for centenarians. It might be good to provide a table of such a composition.

Thank you for the suggestion. We provided a table of the most known changes in the gut microbiota composition of centenarians (Table 1).

-       Line 374 is it not possible that these healthy volunteers seronegative for a specific virus were seropositive for a different virus which shared enough similarities in their protein composition (or peptides structure derived from the virus) to have T-cells which cross-react with another virus?

We rewrote the sentence to better explain the concept (line 435).

-       Line 396 the authors might want to add TLR9 to the text.

We added TLR9 to the text.

We thank You for your constructive critique and we hope the review process has led to an improved manuscript.

If additional changes are warranted, we will make them.

We hope that this revised version of our manuscript may now be found suitable for publication.

Sincerely,

Rossella Cianci, MD, PhD

Round 2

Reviewer 2 Report

Manuscript has been improved.